# Dataset Decomposition: Faster LLM Training with Variable Sequence Length Curriculum

Hadi Pouransari[1,o]    Chun-Liang Li[1]    Jen-Hao Rick Chang[1]
Pavan Kumar Anasosalu Vasu[1]    Cem Koc[1]    Vaishaal Shankar[2,†]    Oncel Tuzel[1]
[1]Apple    [2]Anthropic

## Abstract

Large language models (LLMs) are commonly trained on datasets consisting of fixed-length token sequences. These datasets are created by randomly concatenating documents of various lengths and then chunking them into sequences of a predetermined target length (concat-and-chunk). Recent attention implementations mask cross-document attention, reducing the effective length of a chunk of tokens. Additionally, training on long sequences becomes computationally prohibitive due to the quadratic cost of attention. In this study, we introduce *dataset decomposition*, a novel variable sequence length training technique, to tackle these challenges. We decompose a dataset into a union of buckets, each containing sequences of the same size extracted from a unique document. During training, we use variable sequence length and batch-size, sampling simultaneously from all buckets with a curriculum. In contrast to the concat-and-chunk baseline, which incurs a fixed attention cost at every step of training, our proposed method incurs a computational cost proportional to the actual document lengths at each step, resulting in significant savings in training time. We train an 8k context-length 1B model at the same cost as a 2k context-length model trained with the baseline approach. Experiments on a web-scale corpus demonstrate that our approach significantly enhances performance on standard language evaluations and long-context benchmarks, reaching target accuracy with up to $6\times$ faster training compared to the baseline. Our method not only enables efficient pretraining on long sequences but also scales effectively with dataset size. Lastly, we shed light on a critical yet less studied aspect of training large language models: the distribution and curriculum of sequence lengths, which results in a non-negligible difference in performance.*

## 1  Introduction

Large language models (LLMs) are often pretrained autoregressively (i.e., predicting the next token given a context) on large text corpora sourced from the web. Examples include The Pile [19], RefinedWeb [46], RedPajama [14], and DOLMA [57]. Each of these datasets comprises multiple documents, ranging from Wikipedia articles to books and code repositories. While the individual lengths of the documents vary from a few words (e.g., a message) to hundreds of thousands of words (e.g., a book), the training infrastructure often supports only a limited sequence length in a batch. To facilitate efficient training, document chunking is necessary. In this paper, we investigate the influence of document chunking, propose alternative strategies, and evaluate the proposed strategies with careful experiments.

---

oCorresponding author: mpouransari@apple.com, †Work is done when at Apple.
*Code to be available at `https://github.com/apple/ml-dataset-decomposition`.

38th Conference on Neural Information Processing Systems (NeurIPS 2024).

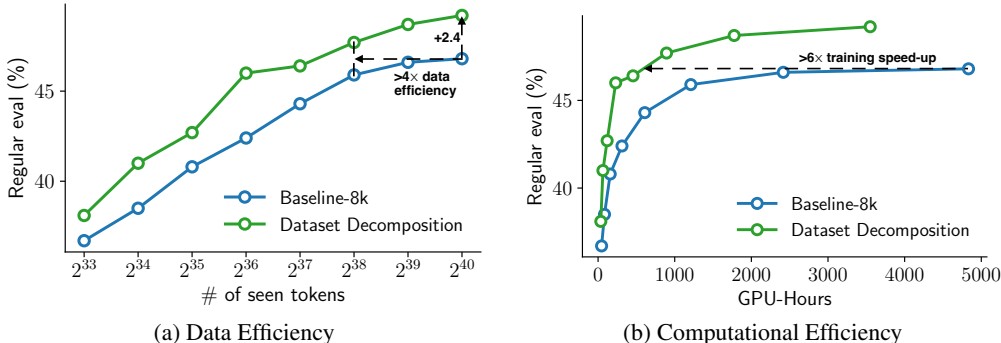

(a) Data Efficiency

(b) Computational Efficiency

Figure 1: Scaling the training of the OpenLM-410M model to 1.1 trillion tokens on the RefinedWeb dataset. Note that each point on the figure represents a separate training from scratch (not different checkpoints of a single run). For the largest run, some tokens are seen more than once (the dataset has approximately 525 billion tokens). For dataset decomposition, we use the Grow-P2 curriculum with 8 cycles. Maximum context length is 8192 and all hyper-parameters are the same for DD and the baseline. **(a)** Regular metrics average when training with the baseline method and the proposed method. We observe more than $4\times$ data efficiency compared to the baseline. Also, even at 1.1 trillion tokens, DD has a +2.4 accuracy improvement compared to the baseline (which has a plateauing accuracy curve even on a logarithmic x-axis). (b) Comparison of model average accuracy versus training cost (GPU-hours). DD reaches the best accuracy of the baseline more than $6\times$ faster. This is the combined effect of DD accuracy and speed gains.

Recent works [43, 37, 59, 60] popularized the *concat-and-chunk* approach to convert text datasets with variable document lengths into sequences with a fixed target length. In this approach, during a data preparation stage before training, we first randomly shuffle and concatenate all tokenized documents. Consecutive concatenated documents are separated by a special token , allowing the model to detect document boundaries. We then chunk the concatenated sequence into subsequences with a *target sequence length*. For example, $2048$ and $4096$ for the Llama-1 and Llama-2 models, respectively. The model is then pretrained on batches of sequences with fixed length.

The concat-and-chunk approach has several shortcomings. First, randomly concatenating documents can lead to the model attending to a context from an unrelated document to predict the next token. While well-trained models learn to avoid cross-document attention, this is not explicitly enforced, leading to potential spurious modeling. Second, the cross-document attention spends unnecessary computation on attending to unrelated tokens that do not facilitate learning. This is especially crucial due to the quadratic complexity of the attention mechanism. Even with an implementation of attention that supports cross-document attention masking, the computational cost for each optimization step would be bottlenecked by the longest document in the global batch, leading to significant under-utilization of devices with shorter documents. Third, even if a document is shorter than the target sequence length, it may still be broken into two chunks when they are at the boundary of two sequences. This results in significantly smaller average chunk lengths compared to the original document length average (see Fig. 3a), which hinders the model's capability.

Recent and concurrent works on LLM training try to improve the concat-and-chunk approach: document-masking is possible with recent implementation of attention [29] as adopted in some recent pre-training recipes [39], best-fit packing [17] to reduce document chunking, and concatenating semantically related documents instead of randomly [55]. However, none of them address all three issues mentioned above together.

In this work, we introduce *dataset decomposition* (DD), a novel approach to decompose data based on their length and train with *variable sequence length* (VSL) and length-based curriculum to address the above issues. We obtain significant both significant accuracy improvement and straining speed-up as shown in Fig. 1. DD decomposes a given dataset containing documents of variable lengths into a union of datasets/buckets, each with sequences of a fixed length. Specifically, a dataset $\mathcal{D}$ is decomposed into buckets $\cup_i \mathcal{D}_i$, where each bucket $\mathcal{D}_i$ contains sequences of length $2^i$, each extracted from a unique document. During training with VSL, at every step of the optimization process, we sample $i$ (based on a curriculum) to form a batch with $b/2^i$ sequences from the bucket $\mathcal{D}_i$, which

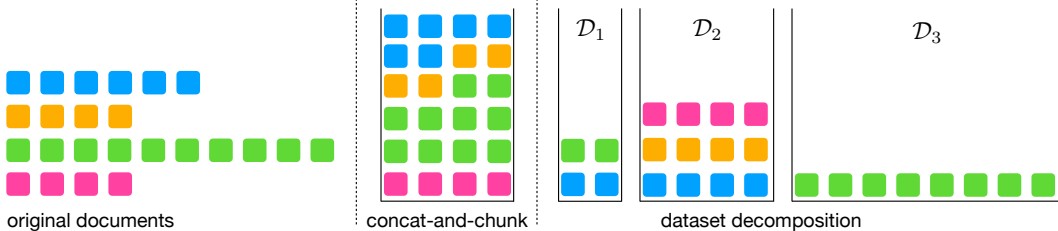

Figure 2: Each cell in the figure represents a token. **Left:** Original documents with variable lengths. **Middle:** Concat-and-chunk baseline to form sequences with a fixed target length (here $= 4$). **Right:** Dataset decomposition method with $\mathcal{D}_1$, $\mathcal{D}_2$, and $\mathcal{D}_3$ buckets .

keeps the total number of tokens in a batch constant ($2^i \times b/2^i = b$), regardless of which $\mathcal{D}_i$ is sampled.

This approach gives us several advantages and resolves the aforementioned issues of the concat-and-chunk method. First, DD is simple and has negligible computational overhead during the data preparation stage, making it easy to scale to large datasets. Second, tokens in each sequence are ensured to be from the same document by construction, which avoids cross-document attention. Furthermore, we have access to the sequence length distribution (an auxiliary prior knowledge) which can be used to create different mixtures/curricula for training. Finally, our VSL training strategy accelerates training time: the latency for one optimization step is less when sampling from $\mathcal{D}_i$ with smaller $i$ (due to attention's quadratic complexity). Following is a summary of our contributions:

- We introduce DD, a method to efficiently decompose a dataset of variable-length documents into a union of buckets with fixed-length sequences. DD enables efficient and robust training via VSL and length-based curriculum.

- We perform large-scale experimentation using different models, datasets, and evaluation tasks to demonstrate the efficacy of the proposed method. We show (see Fig. 1) significant gains in data efficiency ($> 4\times$) and compute efficiency (11% to 45%), resulting in combined LLM pretraining acceleration of up to $6\times$ (time to reach certain accuracy compared to baseline).

- Through careful experimentation, we study the importance of sequence length distribution and mixture during pretraining for different natural language and long-context tasks. We show the effect of concatenation and chunking operations to synthetically alter sequence length (Section 3.2).

## 2 Method

### 2.1 Dataset decomposition

Given a dataset $\mathcal{D}$ of tokenized documents $\{d_1, d_2, \ldots, d_n\}$, the goal of dataset decomposition (DD) is to reorganize $\mathcal{D}$ as a union of buckets, $\cup_i \mathcal{D}_i$, such that: (1) each bucket $\mathcal{D}_i$ consists of sequences of tokens with length $l_i$; (2) each sequence $s \in \mathcal{D}_i$ is a subsequence of one document $d \in \mathcal{D}$; and (3) each token in $\mathcal{D}$ appears in exactly one $\mathcal{D}_i$. This decomposition produces sequences that each belong to a unique document, ensuring no cross-document attention within a sequence during training. Additionally, all sequences in a given bucket $\mathcal{D}_i$ have the same length $l_i$, enabling efficient batching.

Dataset decomposition as defined above is not unique. We propose a specific decomposition, with $l_i = 2^i$, to optimally maintain the original document sequence length distribution while also enabling efficient batch pretraining, as explained in Section 2.2. We apply decomposition at the document level, which makes it very easy to integrate the method into any existing data preparation pipeline (a stage before model training) and is scalable to large datasets. For a tokenized document $d \in \mathcal{D}$ with length $l$, where $l = 2^{i_1} + 2^{i_2} + \ldots + 2^{i_k}$ represents its binary decomposition, we break $d$ into $k$ adjacent sequences $s_1, \ldots, s_k$, with lengths of $2^{i_1}, \ldots, 2^{i_k}$, respectively. Each sequence $s_j$ of length $2^{i_j}$ is then assigned to bucket $\mathcal{D}_{i_j}$. Fig. 2 shows a schematic representation of this method.

With our proposed dataset decomposition approach, each bucket $\mathcal{D}_i$ contains sequences extracted from an original document $d$ such that the length of $d$ is at least $2^i$. In Fig. 3b, we show the distribution

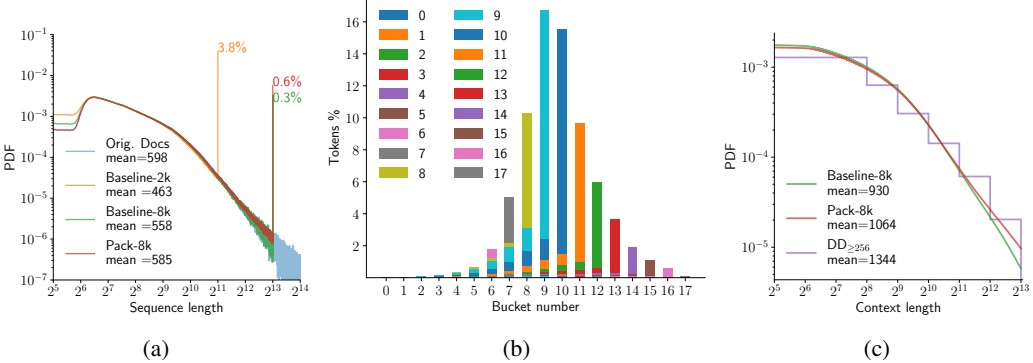

(a)                                        (b)                                        (c)

Figure 3: For the RefinedWeb dataset [46]: **(a)** Distribution of chunk lengths using different dataset preparation methods. Peaks show the percentage of chunks for each method with the same length as the target sequence length. **(b)** Distribution of tokens over $\mathcal{D}_i$'s in DD. Color/pattern shows the $\lfloor \log_2 l \rfloor$, where $l$ is the length of the original document each token is extracted from. **(c)** Probability distribution of context length (number of tokens from the same document a token can attend to) observed during training for the concat-and-chunk baseline with target sequence length 8192 and DD with $\geq 256$ mixture defined in Table 1.

of RefinedWeb dataset tokens over different buckets, where $\mathcal{D}_9$ (corresponding to sequences with length 512) has the maximum tokens. We also highlight the original document lengths from which tokens are extracted. Most tokens in a bucket $\mathcal{D}_i$ are extracted from documents with length $l$ such that $2^i \leq l < 2^{i+1}$, and some tokens are rolled over from documents with length $l \geq 2^{i+1}$. This demonstrates the efficacy of the method in retaining original document length, especially for long documents, which are scarce.

In Fig. 3a, we show the distribution of original document lengths and chunks within 2048 and 8192 target sequence lengths formed by the concat-and-chunk approach. We also present the length distribution using the bin-packing approximate algorithm introduced by a concurrent work [17]. Additionally, in Fig. 3c, we show the distribution of context length (the number of tokens from the same document a token can attend to during pretraining) when using baselines with a target sequence length of 8192 and DD. See Appendix F for additional discussion on sequence length statistics.

In contrast to the concat-and-chunk approach, which results in a static dataset, DD enables us to use sequence length distribution as prior knowledge and optimize the best mixture for the target task. In Section 3.2, we show the bias of each target evaluation toward a sequence length and the effect of concatenation and chunking on model performance. In Section 3.3, we study the effect of different sequence mixtures for LLM pretraining, a less-studied topic in LLM pretraining.

## 2.2 Variable sequence length training

Following the setup in Section 2.1, we assume a set of $k$ buckets such that $\mathcal{D}_i$, containing sequences with length $2^i$, are available. Let $b$ be the target batch size – the number of tokens used per optimization step. In variable sequence length (VSL) training, at every step of optimization, we first sample $i$ from available choices, then pick $b/2^i$ sequences from bucket $\mathcal{D}_i$. Since $\mathcal{D}_i$ consists of sequences with length $2^i$, the number of seen tokens per optimization step remains $b$, independent of the choice of $i$. Training LLMs with the VSL algorithm comes with several advantages.

First, since the total number of seen tokens per optimization step does not change, VSL does not alter optimization dynamics, and the same hyperparameters as the baseline can be utilized (see Section 3).

Second, in Section 3.1, we show that the time to complete one optimization step (forward+backward) for a fixed $b$ (tokens per step) varies by sequence length due to the quadratic cost of attention [63]. With VSL training, the cost of every optimization step depends on the bucket $\mathcal{D}_i$ sampled for that step (and hence the sequence length). Thus, the more expensive steps (corresponding to long sequences) are compensated with less expensive steps (corresponding to short sequences).

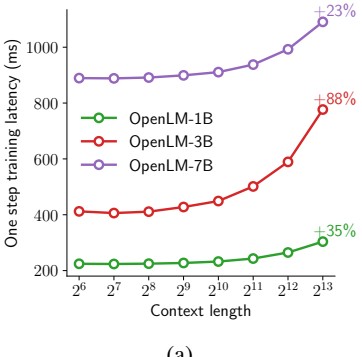
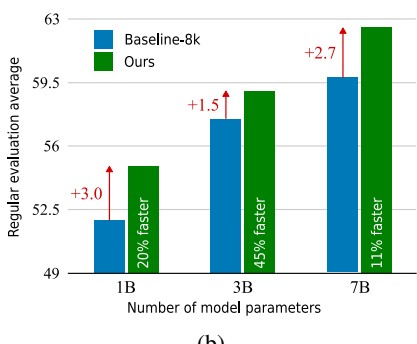

Figure 4: **(a)** Average time for one optimization step ($b = 8 \times 8192$ tokens) on an $8 \times$H100 node with FSDP and FlashAttention2 for different context lengths. **(b)** OpenLM-1B/3B/7B models trained on 137B tokens. Accuracy and training speed gains are shown.

Finally, the sampling component in VSL (which $\mathcal{D}_i$ to choose at every optimization step) enables different curricula of sequence lengths. In Section 3.4, we show the significance of such curricula on model stability and generalization accuracy.

# 3 Experiments and analysis

In this section, we show the efficacy of the proposed method to train LLMs of different sizes on large-scale datasets and provide additional analyses. For all experiments, except the results in Section 3.5, we use RefinedWeb [46] filtering of Common Crawl [1] with a total of $\sim 525$ billion tokens using the EleutherAI/gpt-neox [9] tokenizer (vocabulary size is 50,432). Model architectures and training code are based on the OpenLM [22][†]. For all experiments, other than model scaling in Section 3.5, we use the OpenLM-1B model with an 8k context length. Please refer to Appendix B for implementation details of all experiments.

**Positional encoding** We use Rotary Positional Embedding (RoPE) [58] to encode positions in queries and keys before the attention module. RoPE rotates the consecutive components of queries and keys with a base frequency $f_b = 10,000$. Recent studies [48, 64, 36] have suggested increasing $f_b$ to better adapt a pretrained model for longer sequences through fine-tuning. We find that using a larger $f_b$ is also beneficial when training LLMs from scratch. In Table 4, we show that increasing $f_b$ to 100,000 improves performance for both the baseline and DD methods.

**Evaluation** We evaluate each model on a comprehensive set of standard benchmarks, mainly using LLM Foundry [2]. We report averaged accuracies over each category, as well as the *regular average*, which is the average of 14 regular language modeling benchmarks detailed below:

- **Commonsense Reasoning (CSR)**: PIQA-0-shot [8], COPA-0-shot [52], and OpenBookQA-10-shots [40].
- **Language Understanding (LU)**: Lambada-OpenAI [44], Hellaswag-0-shot [65], Winograd-3-shots [30], and WinoGrande-5-shots [54].
- **Reading Comprehension (RC)**: SQuAD-3-shots [50], BoolQ-0-shot [12], and CoQA-0-shot [51].
- **World Knowledge (WK)**: Jeopardy-3-shots [3], ArcEasy-3-shots [13], ArcChallenge-3-shots [13], and WikiDataQA-3-shots [4]

To evaluate model on longer context tasks, we adopt the following real-world benchmarks:

- **Multi-Document Question Answering (MDQA)**: We follow the exact setup as in Liu et al. [35], where for each question from NaturalQuestions-Open [28, 27], $r$ Wikipedia documents are retrieved such that one of them has the answer to the question, and the other $r - 1$ documents are distractors. We report MDQA-10, MDQA-20, and MDQA-30 accuracy corresponding to $r = 10, 20$, and 30,

---

[†]https://github.com/mlfoundations/open_lm

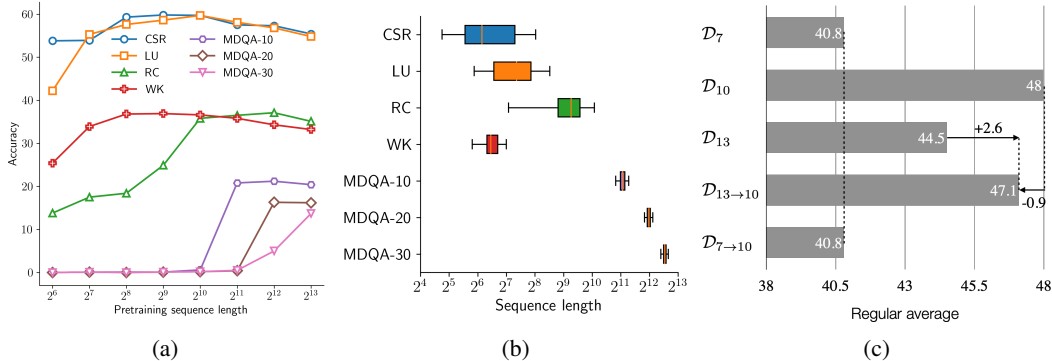

(a)                                       (b)                                       (c)

Figure 5: **(a)** Performance of OpenLM-1B model trained on $2^{34}$ tokens from buckets with different sequence lengths. **(b)** distribution of lengths of documents for different benchmarks. **(c)** Effect of chunking ($\mathcal{D}_{13\to10}$) and concatenating ($\mathcal{D}_{7\to13}$) sequences during pretraining on model performance.

respectively. For each query, we evaluate the model by changing the location of the target document among distractors and report the averaged accuracy.

 • **TOEFL**: This dataset is a multiple-choice question answering dataset from An et al. [5]. The dataset contains QA pairs for 15 longest lectures in Tseng et al. [61], Chung et al. [11]. Only one of the choices is the correct response. We estimate the correct choice by picking the choice with the lowest mean log probability value.

 • **QuALITY**: This dataset is a multiple-choice question answering dataset from An et al. [5]. The dataset contains a long passage for context, followed by a question with multiple choices. Only one of the choices is the correct response. We estimate the correct choice by picking the choice with the lowest mean log probability value.

## 3.1   Training efficiency

We first verify that VSL training enables a higher throughput than the baseline concat-and-chunk method. We enumerate model sizes (OpenLM-1B/3B/7B) and different context lengths ($2^6$ to $2^{13}$) and measure the time to train 100 batches with a fixed global batchsize of $b = 8 \times 8192$ distributed over 8 GPUs in a single node. We repeat this 5 times and report the average time per optimization step in Fig. 4a (with STD mostly $< 1$ms). See Appendix C.1 for additional results with different batchsizes $b$. For each model, we highlight the training time overhead (due to attention's quadratic complexity with an optimized FlashAttention2 kernel [15]) when training with 8192 context lengths compared to 64 context lengths: +35%, +88%, and +23% for OpenLM-1B, -3B[‡], and -7B, respectively. Training overhead grows for longer context lengths (see Fig. 7 for results up to 16k context length).

The concat-and-chunk baseline method always operates at a fixed sequence length. For example, for the OpenLM-1B model, an optimization step with concat-and-chunk takes 243ms and 304ms for target context lengths of 2048 and 8192, respectively. The expected time for VSL, on the other hand, is the weighted average over different sequence lengths depending on the mixture. In Table 1, we report the training step time for different mixtures. For example, with the natural length distribution resulting from DD (Fig. 3b), training up to length 8192 sequences takes a similar time (244ms) as baseline training with length 2048 (with 243ms per step) per step—equivalent to a 20% training time reduction compared to baseline training with a fixed length of 8192 (with 304ms per step).

## 3.2   Sequence length bias

In this section, we study the effect of pretraining data sequence length on model performance in isolation. Using a single bucket $\mathcal{D}_i$ as the dataset, we train an LLM from scratch on sequences with length $2^i$ for a total of $2^{34}$ seen tokens. Note that the number of tokens per optimization step is fixed at 256, irrespective of sequence length. We use the same training hyperparameters for all runs. In Appendix C.2, we show that our conclusions do not depend on the choice of hyperparameters. To

---

[‡]OpenLM-3B has 32 heads ($\times 2$ that of OpenLM-1B), and a per-head dimension of 2560/32=80, not suitable for FlashAttention. This makes attention a significant bottleneck for this model.

| Name | Number of tokens $\times 2^{30}$ | | | | | | | | Avg. Seq. Len | Avg. Ctx. Len | Step Time (ms) | Regular | | | | | MDQA | | | |
|---|---|---|---|---|---|---|---|---|---|---|---|---|---|---|---|---|---|---|---|---|
| | $\mathcal{D}_6$ | $\mathcal{D}_7$ | $\mathcal{D}_8$ | $\mathcal{D}_9$ | $\mathcal{D}_{10}$ | $\mathcal{D}_{11}$ | $\mathcal{D}_{12}$ | $\mathcal{D}_{13}$ | | | | CSR | LU | RC | WK | Avg. | 10 | 20 | 30 | Avg. |
| Natural | 3 | 6 | 10 | 17 | 21 | 17 | 13 | 9 | 482 | 1018 | 244 | 62.4 | 65.4 | 43.8 | 43.9 | **54.0** | **26.7** | 20.7 | **18.5** | 21.9 |
| Equal | 12 | 12 | 12 | 12 | 12 | 12 | 12 | 12 | 257 | 1020 | 244 | 61.9 | 64.3 | 43.1 | 43.5 | 53.3 | 25.1 | 21.4 | 17.4 | 21.3 |
| 1k-only | 0 | 0 | 0 | 0 | 96 | 0 | 0 | 0 | 1024 | 512 | 234 | 60.8 | **66.4** | 43.2 | **44.7** | **54.0** | 0.2 | 0.1 | 0.2 | 0.2 |
| ≤2k | 16 | 16 | 16 | 16 | 16 | 16 | 0 | 0 | 195 | 336 | 231 | **62.8** | 63.7 | 41.8 | 43.5 | 53.1 | 23.5 | 0.4 | 0.4 | 8.1 |
| ≥256 | 0 | 0 | 16 | 16 | 16 | 16 | 16 | 16 | 780 | 1344 | 250 | 61.5 | 65.6 | 43.4 | 44.1 | 53.8 | 25.0 | 18.4 | 17.2 | 20.1 |
| Mid | 0 | 0 | 24 | 24 | 24 | 24 | 0 | 0 | 546 | 480 | 233 | 61.9 | 65.5 | 42.5 | 43.8 | 53.6 | 19.1 | 0.0 | 0.1 | 6.4 |
| ≥1k | 0 | 0 | 0 | 0 | 24 | 24 | 24 | 24 | 2185 | 1920 | 263 | 61.9 | 65.0 | **45.8** | 43.3 | **54.0** | **26.7** | **21.6** | 18.1 | **22.1** |

Table 1: Effect of pretraining **dataset mixture** on model performance. Each row corresponds to a model trained on a specific mixture of dataset decomposition buckets. All models are OpenLM-1B, have seen a total of $96 \times 2^{30}$ tokens, use RoPE with a base frequency of 10k, and are trained with the same hyperparameters. The definition of average context length is given in Appendix F.

reduce statistical error, we train each model twice from scratch with different random seeds and report the average metric for each benchmark (observing an average standard deviation of $\sim 0.3$ for regular benchmarks and $\sim 1.6$ for multi-document QA). Results are demonstrated in Fig. 5a.

We show a significant correlation between pretraining sequence length and different benchmarks. Specifically, the accuracy of commonsense reasoning, language understanding, and world knowledge shows an inverted U-shape behavior with respect to pretraining sequence length, while reading comprehension benefits from longer sequences. This behavior can be associated with training-test distribution alignment with respect to sequence length. In Fig. 5b, we show the length distribution for different benchmarks where RC demonstrates a heavier tail compared to CSR, LU, and WK. Multi-document QA benchmarks show a vivid correlation with respect to sequence length: test accuracy is $\approx 0$ unless pretraining sequence length is greater than the test context length, which is $\sim$ 2k, 4k, and 6k for MDQA-10, -20, and -30, respectively.

It could be argued that data selection based on sequence lengths could introduce bias since the content (or source) of the documents might change based on the sequence lengths. To better understand the effect of sequence length on common metrics, we created two new buckets, $\mathcal{D}_{13 \to 10}$ and $\mathcal{D}_{7 \to 10}$, from existing buckets $\mathcal{D}_{13}$ and $\mathcal{D}_7$, respectively. The bucket $\mathcal{D}_{13 \to 10}$ contains sequences of length $2^{10}$ created by *chunking* sequences from $\mathcal{D}_{13}$ into 8 subsequences and then performing a global shuffle. The bucket $\mathcal{D}_{7 \to 10}$ also includes sequences of length $2^{10}$, each formed by *concatenating* 8 random sequences from $\mathcal{D}_7$.

In Fig. 5c, we compare the regular average metric of models pretrained on these buckets; for each bucket, we train two models from scratch using different random seeds and report the averaged results. $\mathcal{D}_{13 \to 10}$ gains 2.6 points compared to $\mathcal{D}_{13}$ while including the same content. This demonstrates the pure effect of sequence length on model accuracy. Furthermore, training on $\mathcal{D}_{13 \to 10}$ underperforms $\mathcal{D}_{10}$ by 0.9 points, even though they are of the same length, indicating that long documents (used to construct $\mathcal{D}_{13 \to 10}$) correlate less with our benchmarks than short documents (used to construct $\mathcal{D}_{10}$). Finally, we show that *concatenation*, as opposed to *chunking*, does not mitigate length correlation. This is evident from the fact that $\mathcal{D}_{7 \to 10}$ scores the same as $\mathcal{D}_7$ and still significantly worse than $\mathcal{D}_{10}$.

Our analysis suggests that effective base model pretraining requires a mixture of different sequence lengths to perform well on all benchmarks. Next, we systematically study the effect of dataset mixture from the sequence length perspective.

### 3.3 Data mixture

A key benefit of dataset decomposition is access to and control over sequence length distribution. We form datasets with different mixtures of sequence lengths and explore the performance of a model trained on each mixture. Table 1 shows the results. For all experiments, the total seen tokens and hyperparameters are fixed, and only the distribution over sequence length is changed. First, we observe that mixtures with small average context length (we provide the exact definition in Appendix F) perform poorly on MDQA, which requires long context understanding. For example, as for "1k-only", "≤2k", and "Mid" distributions that do not include long sequences from $\mathcal{D}_{12}$ and $\mathcal{D}_{13}$. Larger average context length (e.g., as in "≥1k") also correlates positively with performance on

| Name | Sampling Odds | | | | | | Num. Cycles | Regular | | | | | MDQA | | | |
|---|---|---|---|---|---|---|---|---|---|---|---|---|---|---|---|---|
| | $\mathcal{D}_8$ | $\mathcal{D}_9$ | $\mathcal{D}_{10}$ | $\mathcal{D}_{11}$ | $\mathcal{D}_{12}$ | $\mathcal{D}_{13}$ | | CSR | LU | RC | WK | Avg. | 10 | 20 | 30 | Avg. |
| Uniform | 1 | 1 | 1 | 1 | 1 | 1 | 1 | 62.2 | 65.2 | 43.4 | 44.0 | 53.8 | 27.3 | 22.0 | 19.6 | 23.0 |
| Grow-Linear | 6 | 5 | 4 | 3 | 2 | 1 | 1 | 60.9 | 64.2 | 46.6 | 42.9 | 53.6 | 30.9 | 26.0 | 23.9 | 26.9 |
| | | | | | | | 8 | 62.7 | 65.0 | 45.4 | 44.7 | 54.5 | 30.1 | 25.3 | 22.8 | 26.1 |
| Grow-P2 | 32 | 16 | 8 | 4 | 2 | 1 | 1 | 60.9 | 64.3 | 46.5 | 44.1 | 54.0 | 29.6 | 25.0 | 23.1 | 25.9 |
| | | | | | | | 8 | 62.8 | 65.2 | 45.3 | 44.2 | 54.4 | 32.3 | 26.9 | 24.6 | 28.0 |
| Grow-P100 | $100^5$ | $100^4$ | $100^3$ | $100^2$ | 100 | 1 | 1 | 60.8 | 65.0 | 47.3 | 43.4 | 54.1 | 30.6 | 26.9 | 23.5 | 27.0 |
| | | | | | | | 8 | 63.2 | 65.4 | 46.3 | 44.6 | 54.9 | 30.2 | 23.2 | 18.9 | 24.1 |
| Shrink-P100 | 1 | 100 | $100^2$ | $100^3$ | $100^4$ | $100^5$ | 1 | 60.0 | 62.2 | 37.6 | 40.7 | 50.3 | 24.5 | 18.7 | 15.6 | 19.6 |

Table 2: Effect of **length-based curriculum**. All models are OpenLM-1B and have seen a total of $96 \times 2^{30}$ tokens, with exactly $2^{34}$ tokens from each $\mathcal{D}_i$ for $i = 8, \ldots, 13$. We use RoPE with a base frequency of 100k and the same default hyperparameters.

reading comprehension tasks, consistent with our observation in Fig. 5a, but comes at the cost of a longer training step time.

Furthermore, "1k-only", that is training using only the best sequence length ($= 1024$) from the study in Section 3.2 results in good performance on regular evaluations, especially for language understanding and world knowledge tasks, but is poor for long context tasks. Finally, we observe that "natural" mixture, that is aligned with the distribution resulting from dataset decomposition (see Fig. 3b), obtains near-optimal performance on both regular and MDQA tasks, demonstrating the scalability of the proposed approach to large datasets without a need for intervention on the natural underlying length distribution.

## 3.4 Length-based curriculum

We can think of short sequences as being "easier" compared to longer ones; hence motivating a curriculum learning [7, 18] that prioritizes short sequences. A similar idea (training with image resolutions from low to high) is explored in vision to train CLIP [49] models more efficiently [33]. In VSL, we can easily implement curriculum learning through sampling designs. At every optimization step, we sample *without replacement* a batch with $b$ tokens from bucket $\mathcal{D}_i$ with probability $p_i$. If a bucket is empty, we exclude it from sampling. We study different curricula for the "$\geq 256$" mixture (with an equal number of tokens in $\mathcal{D}_8, \ldots, \mathcal{D}_{13}$). Results are shown in Table 2. For each curriculum, we determine the odds of picking a batch from each bucket ($= p_i$'s when normalized). Details of our length-based sampling and curriculum are provided in Algorithm 1. We consider curricula that shift from short to long sequences at different paces controlled by $p_i$'s changing linearly, with powers of 2, and with powers of 100 between buckets.

Due to the presence of other hyperparameter schedules during the course of training (e.g., learning rate and weight decay), a curriculum on length may result in a potential implicit bias. For example, if we only see long sequences toward the end of training, long sequence learning occurs only when the learning rate is too small. To address this potential issue, we also explore cyclic curricula, where a curriculum is applied in cycles similar to cyclic learning rate schedules [56] as shown in Fig. 6. Note that when we train on a sequence of length $l$, we have $l$ next-token prediction losses (applied in parallel) with context lengths $0, 1, \ldots, l - 1$. This already implies some mixing: when training on a "hard" example (i.e., a long sequence), we also include "easy" examples (its shorter sub-sequences). Therefore, even towards the end of each cycle, we still have some losses with short contexts.

Our results show that the cyclic "Grow-P2" curriculum is near optimal with different metrics. An additional benefit of curriculum is training stability. Li et al. [31] noticed that long sequences contribute to extreme gradient variance, especially at the beginning of training, resulting in instability. We also observe (see Appendix E) that our proposed approach with curriculum results in more stable training dynamics, thus enabling more efficient training with larger batch sizes and learning rates.

## 3.5 Scaling

**Dataset scaling** In Fig. 1a, we show the performance of models trained with $2^{34}, 2^{35}, 2^{36}, 2^{37}$, and $2^{38}$ total tokens using DD and baseline. We use the "$\geq 256$" mixture and "Grow-Linear" curriculum with 8 cycles for DD, and a fixed target sequence length 8192 for the baseline. Results show $> 2\times$ data efficiency: our proposed method reaches the same accuracy as the baseline using less than half the tokens.

| Model Size | Method | Num GPUs | Time (hours) | Δ | Regular Avg. | Δ | MDQA Avg. | Δ |
|---|---|---|---|---|---|---|---|---|
| 160M | Baseline-8k | 16 | 18.3 | - | 39.3 | - | 9.7 | - |
| | DD | | 15.7 | -14% | 40.0 | +0.7 | 11.4 | +1.7 |
| 410M | Baseline-8k | 16 | 38.9 | - | 48.3 | - | 14.8 | - |
| | DD | | 29.6 | -24% | 49.4 | +1.1 | 18.8 | +4.0 |
| 1B | Baseline-8k | 32 | 44.4 | - | 56.7 | - | 25.6 | - |
| | DD | | 35.4 | -20% | 58.4 | +1.7 | 25.6 | - |

Table 3: Comparing baseline training with DD on an alternative pretraining dataset and model sizes.

| Method | $f_b$ | Regular Avg. | MDQA Avg. |
|---|---|---|---|
| Baseline-8k | 10k | 51.3 | 19.0 |
| | 100k | 51.5 | **24.4** |
| $DD_{\geq 256}$ | 10k | 53.8 | 20.1 |
| | 100k | 53.8 | **24.9** |

Table 4: Effect of RoPE base frequency, $f_b$, in pretraining.

**Model scaling**   We report results on OpenLM-1B, -3B, and -7B trained from scratch for a total of $2^{37}$ tokens in Fig. 4b. We compare baseline training with a fixed target sequence length 8192 and VSL training with a $DD_{\geq 256}$ mixture and the "Grow-Linear" curriculum with 8 cycles. Training with DD results in significant accuracy gains and reductions in training wall-clock time at different scales.

**Alternative dataset**   We demonstrate the efficacy of our proposed method on another large-scale dataset, DataComp-LM [32]. We train models with different numbers of parameters: OpenLM-160M, -410M, and -1B, for a total of 137B tokens. We compare the baseline with a $DD_{\geq 256}$ mixture trained with the "Grow-P2" curriculum with 8 cycles. Results are reported in Table 3, demonstrating significant accuracy and training efficiency gains.

## 3.6   Comparison with state-of-the-art

We compare our proposed method, data decomposition, with other approaches for handling various document lengths of pretraining data, including document masking (DM), best-fit sequence packing [17], and in-context pretraining (ICLM) [55]. We describe the details of our implementation of the best-fit packing in Appendix D. For ICLM, we use the official implementation[§] applied to the RefinedWeb dataset. The results are shown in Table 5.

Pre-training context length is an important factor in determining a model's long-context performance. We empirically validate this in the results shown in Fig. 5a, where models trained on longer sequences perform better on multi-document QA. Our proposed method has an average context length (as defined in Eq. (2)) of 1,344 for the RefinedWeb dataset, compared to 930 for the baseline (see Fig. 3c) and 1,064 when packing [17] is applied. This explains why the dataset decomposition mixture, even without any length-based curriculum (the first row in Table 2), outperforms Baseline-8k-DM and Pack-8k+DM (second and third rows in Table 5). Here, DM refers to applying document masking during training to avoid cross-document attention.

Document masking improves the baseline on regular evaluations from $51.5$ to $52.4$ by preventing cross-document attention. However, Xiong et al. [64] demonstrate that including concatenated unrelated documents can still enhance long-context metrics compared to training solely with shorter sequences. Therefore, DM experiences a slight decline in long-context evaluations, dropping from $27.5$ to $27.1$. Baseline-8k multi-document QA performance is even slightly better than our proposed dataset decomposition mixture when used *without* length-based curriculum (the first row in Table 2).

In-context pre-training LMs (ICLM) [55] proposes document sorting based on content similarity. Although the benefits of ICLM with large-scale Common Crawl data (used in our experiments) are marginal in regular evaluation, we observe that ICLM results in slightly better multi-document QA performance than Baseline-8k when 30 documents are in the context compared with Baseline-8k ($22.0\%$ vs. $20.5\%$). The average long-context metric boosts from $27.5$ for Baseline-8k to $28.7$ for ICLM. However, the similarity finding step proposed by ICLM is resource-intensive at scale[¶].

Finally, as shown in in Table 2 our proposed cyclic length-based curriculum, for example, Grow-P2 with 8 cycles, results in a significant improvement in the model's long-context capability. Our proposed method avoids cross-document attention to unrelated content, maintains coherent long sequences, and benefits from a length-based curriculum, effectively improving performance in both regular and long-context evaluations compared to all baselines. We further summarize long-context performance of different methods discussed above in Table 6.

---

[§]https://github.com/swj0419/in-context-pretraining

[¶]Processing 400B tokens with the official repository required over a week using 96 CPUs and 16 GPUs.

| Method | Regular | | | | | Long Context | | | | | | Step Time (ms) | Data Prep. Cost |
|---|---|---|---|---|---|---|---|---|---|---|---|---|---|
| | CSR | LU | RC | WK | Avg. | MDQA | | | TOEFL | QuALITY | Avg. | | |
| | | | | | | 10 | 20 | 30 | | | | | |
| Baseline-8k | 60.6 | 62.5 | 41.5 | 41.3 | 51.5 | 29.0 | 23.8 | 20.5 | 26.2 | 32.0 | 27.5 | 304 | $ |
| Baseline-8k+DM | 60.2 | 64.1 | 42.8 | 41.8 | 52.4 | 24.4 | 20.0 | 16.0 | 29.2 | 32.0 | 27.1 | 304 | $ |
| Pack-8k+DM [17] | 60.3 | 64.0 | 44.6 | 41.8 | 52.7 | 25.6 | 19.8 | 16.9 | 29.2 | 33.1 | 27.7 | 304 | $$ |
| ICLM [55] | 60.6 | 62.1 | 44.7 | 40.0 | 51.7 | 26.7 | 20.0 | 22.0 | 28.7 | 34.6 | 28.7 | 304 | $$$ |
| **DD (ours)** | **62.8** | **65.2** | **45.3** | **44.2** | **54.4** | **32.3** | **26.9** | **24.6** | **30.7** | 34.2 | **30.9** | 244 | $ |

Table 5: Comparison with baseline and state-of-the-art methods. All models are trained with the same hyperparameters, RoPE with $f_b = 100k$, and for 103B tokens. DM denotes training with document masking. DD uses the "Grow-P2" curriculum with 8 cycles. Dataset preparation cost is symbolic to compare methods and does not reflect the wall-clock time.

# 4 Related works

Recent works have raised concerns regarding cross-document attention. For example, Llama-3 [39], ICLM [55], and [17], which we discussed in Section 3.6. Similarly, [26] discuss challenges with the baseline concat-and-chunk approach and propose an approximate bin-packing algorithm.

| Method | Doc Masking | Average Context | Docs in a Sequence | Curr. | MDQA-30 (%) |
|---|---|---|---|---|---|
| Baseline | ✓ | 930 | Mult-random | ✗ | 16.0 |
| Pack-8k+DM [17] | ✓ | 1064 | Mult-packing | ✗ | 16.9 |
| DD-Uniform | ✗ | 1344 | Single | ✗ | 19.6 |
| Baseline | ✗ | 4096 | Mult-random | ✗ | 20.5 |
| ICLM [55] | ✗ | 4096 | Mult-semantic | ✗ | 22.0 |
| **DD-Grow-P2** | N/A | 1344 | Single | ✓ | **24.6** |

Table 6: Summary of long-context performance for different methods from Table 2 and Table 5.

Related to our study on sequence length bias, [62] shows the importance of train-vs-test time distribution shift from a sequence length perspective on a string editing task. [6, 66, 25, 36] highlight the challenge of generalizing to lengths beyond what the model has seen during training and discuss the importance of positional encoding. Several works [41, 67, 23, 64, 10, 47, 48, 53, 34] address enabling LLM inference with long context (see [45] for an overview). These approaches are orthogonal to our contribution and can be applied post-pretraining to adapt to longer lengths. GrowLength [24] proposes accelerating LLM pretraining by progressively growing context length using the baseline sequence formation method, but does not show results on LLMs. Similarly, increasing sequence length has been shown in BERT model training [42] to improve compute efficiency.

The idea of dynamic batching has been explored in other domains. In vision, methods like NaViT [16, 38] use images with variable resolutions (a similar concept to context length for LLMs). In seq-to-seq tasks (e.g., automatic speech recognition, text-to-speech, and neural machine translation), the inputs have different lengths. An efficient approach is to sort inputs by their length and form batches of inputs with similar lengths during training (after possible padding). Batchsize is dynamically adjusted inversely proportional to input lengths [20, 21]. Different from these works, in dataset decomposition, we do not simply put documents with similar lengths into the same bucket. Instead, we decompose each document into multiple subsequences and form multiple buckets. We form batches with different lengths during training by sampling from these buckets using a target mixture and curriculum.

# 5 Conclusion and limitations

In this paper, we explore the shortcomings of a popular LLM pretraining approach, concat-and-chunk, and introduce dataset decomposition, a method to decompose a dataset of text documents into buckets containing fixed sequence lengths. We show results of variable sequence training using DD with different mixtures, curricula, datasets, and models, demonstrating significant LLM pretraining speedup and a final model accuracy boost on a wide range of benchmarks. Furthermore, we provide analysis on sequence length bias and attention masking. We compare our proposed method with recent works that also address concat-and-chunk shortcomings in a unified experimental setup and show gains in data preparation cost, training time, and final model accuracy.

**Limitations.** The training speed gains compared to the baseline are significant only when the target sequence length is long enough. Otherwise, the attention cost is not a dominant fraction of training, and hence no significant training speedup is expected.

## Acknowledgements

We thank Tatiana Likhomanenko, Jason Ramapuram, Alexander Toshev, Barry Theobald, and Fartash Faghri from Apple for their valuable feedback and suggestions.

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

# A  Broader impacts

This work enables faster training of LLMs, which are among the most compute-intensive applications in the field. A positive societal/environmental impact of this work is training LLMs with a smaller carbon footprint.

Another potential societal advantage of this work is training LLMs with fewer hallucinations. While we did not directly measure this potential benefit, a concurrent work [17] shows such a benefit when cross-document attention is not allowed during LLM pretraining.

# B  Implementation details

## B.1  Training details

**Software and hardware details**   All experiments in this paper are conducted using the OpenLM[‖] repository, which is based on PyTorch. We use Fully Sharded Data Parallelism (FSDP) with Bfloat16 mixed precision for all experiments. We use the Xformers [29] implementation for attention. For hardware, we use one or more nodes of $8\times$ NVIDIA H100 GPUs (Hopper architecture), each with 80GB memory, and 192 CPU cores with 2000GB of RAM. Nodes are connected through Elastic Fabric Adapter (EFA) for efficient inter-node communication hosted by AWS.

**Model architecture details**   We provide details of all architectures used in the paper in Table 7 to Table 11.

| Model Name | OpenLM-160M |
|---|---|
| Hidden dimension | 768 |
| Number of Layers | 12 |
| Number of Heads | 12 |
| Number of Parameters | 162,435,840 |

Table 7: OpenLM-160M.

| Model Name | OpenLM-410M |
|---|---|
| Hidden dimension | 1024 |
| Number of Layers | 24 |
| Number of Heads | 16 |
| Number of Parameters | 411,665,408 |

Table 8: OpenLM-410M.

| Model Name | OpenLM-1B |
|---|---|
| Hidden dimension | 2048 |
| Number of Layers | 24 |
| Number of Heads | 16 |
| Number of Parameters | 1,439,893,504 |

Table 9: OpenLM-1B.

| Model Name | OpenLM-3B |
|---|---|
| Hidden dimension | 2560 |
| Number of Layers | 32 |
| Number of Heads | 32 |
| Number of Parameters | 2,796,096,000 |

Table 10: OpenLM-3B.

| Model Name | OpenLM-7B |
|---|---|
| Hidden dimension | 4096 |
| Number of Layers | 32 |
| Number of Heads | 32 |
| Number of Parameters | 6,889,672,704 |

Table 11: OpenLM-7B.

**Baseline hyper parameters**   We list our baseline hyperparameters in Table 12 and iterate over changes for each section next. Note that we did not explicitly optimize hyperparameters for any of the experiments, and we always use the same hyperparameters when using either the baseline method or ours.

---

[‖] https://github.com/mlfoundations/open_lm

| Optimizer | AdamW |
|---|---|
| **AdamW-**$\beta_1$ | 0.9 |
| **AdamW-**$\beta_2$ | 0.95 |
| **learning-rate schedule** | cosine+warmup |
| **Maximum learning rate** | $3 \times 10^{-3}$ |
| **cooldown learning rate** | $3 \times 10^{-5}$ |
| **Warm-up steps** | 5000 |
| **Grad Norm Clipping** | 1.0 |
| **Global batchsize (num tokens per step)** | $2^{19}$ |
| **Weight Decay** | 0.1 |
| **Z-Loss Coefficient** | $10^{-4}$ |

Table 12: Baseline hyper-parameters.

**Implementation details of Section 3.2 experiments**    Experiments in this section are done using the same hyperparameters as in Table 12 for a total of $2^{34}$ tokens on the OpenLM-1B model. We trained each model twice with different random seeds and report the averaged results. For models in this section, we use RoPE with $f_b = 10,000$. In Table 12, we show that our results and conclusions in this section are not sensitive to hyperparameters, including the RoPE base frequency $f_b$.

**Implementation details of Section 3.3 and Section 3.4 experiments**    Experiments in this section are done with OpenLM-1B model, trained for total of $96 \times 10^{34} \approx 103$B tokens. Hyper-parameters are the same as Table 12, except we used 20000 warmup steps for all models presented in this section. We use RoPE with $f_b = 10,000$ for all models in Section 3.3 and $f_b = 100,000$ for models in Section 3.4.

**Implementation details of Section 3.5**
**Dataset scaling**: Experiments in this section are trained with the OpenLM-1B model, RoPE with $f_b = 100,000$, and the baseline setup as in Table 12 except for the following changes for different dataset sizes:

- total tokens = $2^{34}$, warmup steps = $5,000$
- total tokens = $2^{35}$, warmup steps = $5,000$
- total tokens = $2^{36}$, warmup steps = $10,000$
- total tokens = $2^{37}$, warmup steps = $20,000$
- total tokens = $2^{38}$, warmup steps = $40,000$

**Model scaling**: Experiments in this section are trained with the OpenLM-1B, OpenLM-3B, and OpenLM-7B models, $2^{37} \approx 137$B total seen tokens, RoPE with $f_b = 100,000$, and the baseline setup as in Table 12 except for the following changes for different model sizes:

- OpenLM-1B, warmup steps = $20,000$, max-lr = $3 \times 10^{-3}$, batchsize $b = 2^{19}$, with 32 H100 GPUs
- OpenLM-3B, warmup steps = $20,000$, max-lr = $2 \times 10^{-3}$, batchsize $b = 2^{20}$, with 64 H100 GPUs
- OpenLM-7B, warmup steps = $20,000$, max-lr = $1 \times 10^{-3}$, batchsize $b = 2^{22}$, with 128 H100 GPUs

**Alternative dataset**: Experiments in this section are trained with the OpenLM-160M, OpenLM-410M, and OpenLM-1B models, $2^{37} \approx 137$B total seen tokens, RoPE with $f_b = 100,000$, and the baseline setup as in Table 12 except for the following changes for different model sizes:

- OpenLM-160M, warmup steps = $20,000$, max-lr = $5 \times 10^{-3}$, weight-decay = 0.033, with 16 H100 GPUs
- OpenLM-410M, warmup steps = $20,000$, max-lr = $4 \times 10^{-3}$, weight-decay = 0.066, with 16 H100 GPUs
- OpenLM-1B, warmup steps = $20,000$, max-lr = $3 \times 10^{-3}$, weight-decay = 0.1, with 32 H100 GPUs

For the DD experiments we used "Grow-P2" length curriculum which is visualized in Fig. 6.

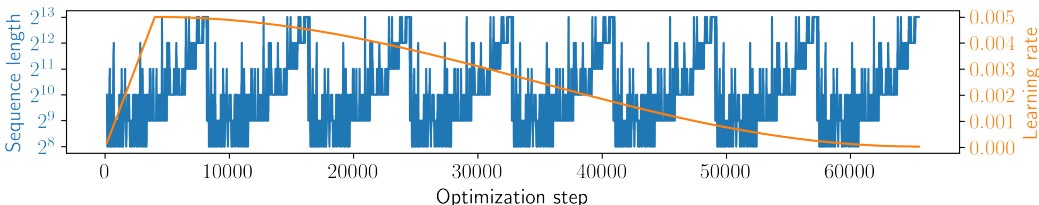

Figure 6: Comparison of length-based curriculum schedule with learning rate schedule. Sequence length varies between 256 and 8192 based on the Grow-P2 curriculum with 8 cycles. Note that the choice of bucket (and hence the sequence length) is random, with sampling probabilities determined by the curriculum. In the figure, we show the length of the sampled sequence at every 9 optimization steps. For the learning rate, we use a cosine learning rate with a warm-up for 4k steps. The job corresponds to training for a total of $2^{36}$ tokens, with $2^{20}$ tokens seen per optimization step.

**Implementation details of Section 3.6** All experiments in this section are done with the OpenLM-1B model, trained for a total of $96 \times 10^{34} \approx 103B$ tokens. Hyperparameters are the same as in Table 12, except we used 20,000 warmup steps for all models presented in this section. We use RoPE with $f_b = 100,000$.

## B.2 Length based sampling and curriculum algorithm

We present the details of our length-based sampling and curriculum in Algorithm 1.

---

**Algorithm 1** Length based sampling and curriculum

---

**Require:**
- $D_i$: list of buckets such that $D_i$ includes sequences with length $2^i$
- $n_i$: total number of tokens to be picked from each bucket (see Table 1)
- $o_i$: sampling odd for each bucket (see Table 2)
- $c$: number of cycles
- $b$: number of tokens per optimization step

$s_{i,j} \leftarrow$ `random subset of` $D_i$ `with` $n_i/c$ `tokens`          $\triangleright$ non-overlapping subsets of $D_i$
**for** $j \in [1, 2, \ldots, c]$ **do**                                          $\triangleright$ loop over cycles
    **while** `at least one` $s_{i,j}$ `is non-empty` **do**
        $odds \leftarrow [o_i$ `if` $s_{i,j}$ `is not empty else` $0$ `for` $i = 1, 2, 3, \ldots]$
        $probs \leftarrow odds/odds.sum()$
        `randomly sample index` $i$ `with probability` $probs[i]$
        `sample` $b/2^i$ `sequences from` $s_{i,j}$ `w/o replacement for training`
    **end while**
**end for**

---

## B.3 Evaluation details

**Multi Document Question Answering (MDQA)** We follow the open-book evaluation setup described in [35]. The document containing the answer is part of the context. The evaluation script provided by the official repository processes the model's response by using only the text before the first occurrence of a newline character as the answer. We noticed that sometimes the model responds with multiple newline characters before providing any valid text. In view of this behavior, we updated the evaluation script to look for the first non-empty text output from the model instead of the first string after newline character. Apart from this change in processing the model output, the rest of the evaluation follows the official implementation [35].

**TOEFL** We follow the setup described in [5]. As described in Section 3, the dataset contains multiple-choice QA pairs for the 15 longest lectures in [61, 11]. To obtain a response from the model, we follow MMLU-style prompting, where the choices are appended to the original prompt individually and the mean log-probability is computed for each choice. The choice corresponding to

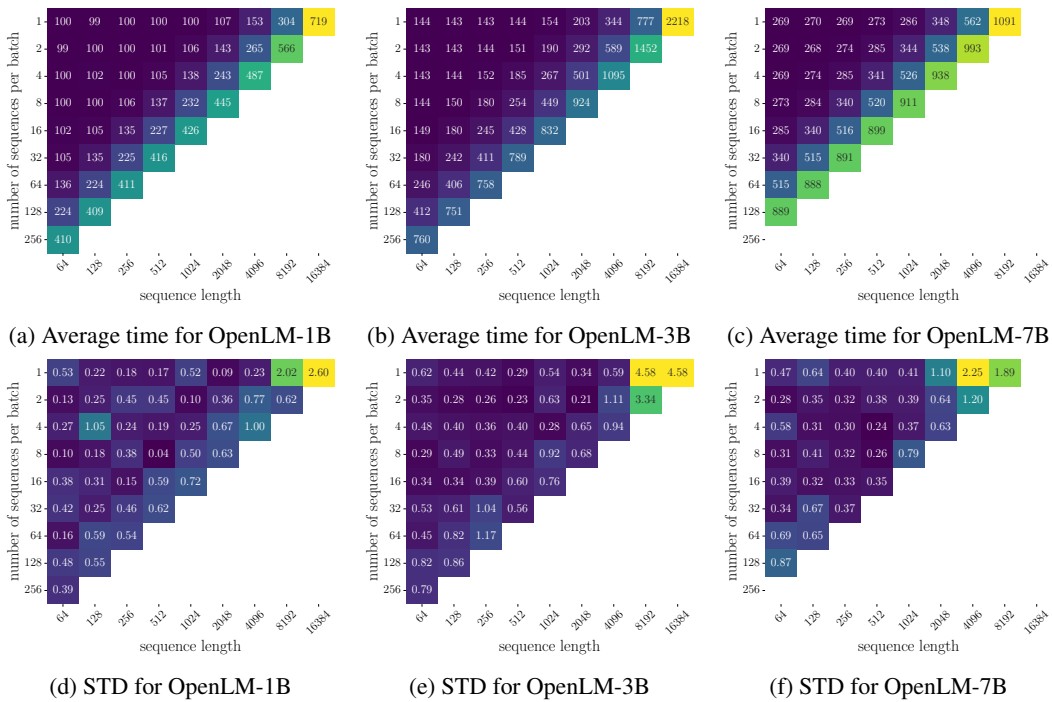

(a) Average time for OpenLM-1B     (b) Average time for OpenLM-3B     (c) Average time for OpenLM-7B

(d) STD for OpenLM-1B     (e) STD for OpenLM-3B     (f) STD for OpenLM-7B

Figure 7: **Top row:** Average time (ms) for each node to train one batch on a $8{\times}$H100 machine using FSDP. **Bottom row:** measured standard deviation for each setup.

the `argmax` of mean log-probability is then chosen as the model's response. After we obtain the response, the computation of accuracy follows the official implementation [5].

**QuALITY** We follow the setup described in [5]. The dataset contains long documents with each document containing multiple-choice QA pairs. Sometimes the context for a QA pair can be longer than 8192 tokens. To account for the longer sequence length, we increase the base frequency of RoPE positional encoding from 100k to 200k without any fine-tuning. To obtain a response from the model, we follow MMLU-style prompting, where the choices are appended to the original prompt individually and the mean log-probability is computed for each choice. The choice corresponding to the `argmax` of mean log-probability is then chosen as the model's response. After we obtain the model output, the rest of the evaluation follows the official implementation [5].

## C   Additional results

### C.1   Additional results for training efficiency

We enumerate model sizes (OpenLM-1B, OpenLM-3B, OpenLM-7B), the number of sequences in a batch (from 1 to 256), and sequence lengths ($2^6$ to $2^{14}$) and measure the time to train 100 batches. We repeat this 5 times and report the average and standard deviation time per batch in Fig. 7. Notice that in the figure, each diagonal corresponds to a fixed $b$ (number of tokens seen per optimization step).

### C.2   Additional results for sequence length bias experiments

In this section, we show that changing hyperparameters does not alter our conclusions in Section 3.2. We observed that pretraining on a sequence length of 1024 results in optimal performance with respect to regular metrics, compared to both longer and shorter lengths. For example, the regular average metric is 48.0 when pretraining with a 1024 sequence length, but it is 47.0 when pretraining with a 2048 sequence length. We explore whether this gap can be filled by using potentially better hyperparameters when training with a 2048 sequence length. Results are shown in Table 13,

demonstrating that the gap cannot be simply filled by choosing a different hyperparameter and is fundamental to the choice of pretraining sequence length.

| Maximum Learning Rate | RoPE $f_b$ | Regular Average |
|---|---|---|
| $3 \times 10^{-3}$ | 10,000 | 47.0 |
| $3 \times 10^{-3}$ | 100,000 | 47.1 |
| $10^{-3}$ | 10,000 | 45.9 |
| $10^{-2}$ | 10,000 | 46.5 |

Table 13: Sensitivity to hyperparameters for Section 3.2 experiments. All models are trained twice with different random seeds, and averaged results are reported.

## C.3 Additional results for scaling experiments

In this section, we show additional results for the experiments presented in Section 3.5. Table 14 shows results for dataset scaling, Table 15 for model scaling, and Table 16 for experiments on an alternative dataset.

| Seen tokens | Method | Regular average | MDQA average |
|---|---|---|---|
| $2^{34}$ | Baseline-8k | 45.2 | 7.8 |
| | DD | 47.0 | 16.0 |
| $2^{35}$ | Baseline-8k | 47.6 | 15.4 |
| | DD | 50.6 | 23.3 |
| $2^{36}$ | Baseline-8k | 50.2 | 19.9 |
| | DD | 52.1 | 22.3 |
| $2^{37}$ | Baseline-8k | 51.9 | 23.2 |
| | DD | 54.9 | 25.9 |
| $2^{38}$ | Baseline-8k | 53.6 | 25.8 |
| | DD | 56.0 | 29.4 |

Table 14: Dataset scaling for OpenLM-1B.

| Model size | Method | Regular average | MDQA average |
|---|---|---|---|
| 1B | Baseline-8k | 51.9 | 23.1 |
| | DD | 54.9 | 24.2 |
| 3B | Baseline-8k | 57.5 | 17.8 |
| | DD | 59.0 | 31.1 |
| 7B | Baseline-8k | 59.8 | 31.7 |
| | DD | 62.5 | 34.7 |

Table 15: Model scaling for total of 137B tokens.

| Size | Method | PIQA 0-shot | COPA 0-shot | OBQA 10-shots | LamOAI 0-shot | HelSwg 0-shot | WinG 3-shots | WinGE 5-shots | SQuaAD 3-shots | BoolQ 0-shot | CoQA 0-shot | Jeop 3-shots | ArcE 3-shots | ArcC 3-shots | WikiQA 3-shots | MDQA 10 | 20 | 30 |
|---|---|---|---|---|---|---|---|---|---|---|---|---|---|---|---|---|---|---|
| 160M | Baseline | 66.5 | 61 | 29.2 | 40.5 | 37.2 | 63.4 | 51.9 | 12.9 | 55.6 | 18.2 | 2.3 | 49.5 | 25.9 | 36.2 | 12.8 | 9.3 | 7.1 |
| | DD | 66.4 | 66 | 30.2 | 43.6 | 37.7 | 66.3 | 52.2 | 14.3 | 50.7 | 19.1 | 4.3 | 51.5 | 24 | 34 | 16.2 | 9.9 | 8.2 |
| 410M | Baseline | 69.8 | 68 | 37.4 | 53.0 | 50.4 | 74.0 | 55.8 | 30.0 | 59.7 | 28.5 | 12.1 | 59 | 29.8 | 48.3 | 18.9 | 13.4 | 12 |
| | DD | 71.5 | 70 | 38 | 55.8 | 51.6 | 74.7 | 56.3 | 27 | 59.5 | 26.2 | 17.6 | 60.4 | 30.5 | 52.2 | 24.4 | 18.1 | 14 |
| 1B | Baseline | 74.9 | 74 | 43.4 | 63 | 62.7 | 80.2 | 63.4 | 41.8 | 64.1 | 35.3 | 29.7 | 65.7 | 38.4 | 56.7 | 31.3 | 24.8 | 20.8 |
| | DD | 76.7 | 75 | 42.6 | 64.7 | 64.7 | 82.8 | 65 | 41.8 | 66.4 | 38.3 | 32.6 | 68.4 | 39.8 | 58.7 | 31.6 | 24.7 | 20.4 |

Table 16: Small model performance trained on an improved refined-web pipeline applied to Common Crawl. All models are trained for a total of $2^{37}$ tokens.

## D Comparison to best-fit sequence packing

Some recent works have employed a bin packing-based strategy [17] which aims to reduce document cross-attention by minimizing unnecessary document truncation. To achieve this, they implement a known approximation algorithm called best-fit decreasing, which packs document chunks into sequences as tightly as possible. To compare with our method, we created a new dataset based on our implementation of the best-fit decreasing algorithm and trained a new model using this dataset. We present our implementation of the best-fit decreasing algorithm, the dataset we created, and the model we trained for comparison.

Given a dataset $\mathcal{D}$, the input to the algorithm is a list of *tokenized* document chunks $C = \{c_1, c_2, \ldots, c_K\}$ such that $\bigcup_{i=1}^{K} c_i = \mathcal{D}$, where each chunk is at most context size $n$ (e.g., 2048) in length. The output of the algorithm is a list of bins $\mathcal{B} = \{b_1, b_2, \ldots, b_M\}$ such that $c_i \in b_j$. As a pre-processing step, we first tokenize the documents and convert them into chunks. Truncation is applied during this step only when necessary. Next, we sort the chunks from largest to smallest and start from the first chunk to pack into bins of size $n$. We track the remaining capacities for each

bin while we iterate over the chunks. In each iteration, the algorithm finds the best bin that is both feasible and optimal for placing the chunk. Feasible bins are those that can accommodate the chunk, and optimal bins are those left with the minimum remaining capacity after placing the chunk. If such a bin is not found, we open a new bin and place the chunk inside. After all the chunks have been placed, we select the bins that have non-zero remaining capacities and fill them with pad tokens `<PAD>`.

We process the RefinedWeb [46] dataset using the aforementioned procedure and create training sequences by concatenating all chunks in a bin. Figure 3 shows that while best-fit packing results in a higher average context length compared to the baseline *concat-and-chunk*, it is still much lower compared to our method *dataset decomposition*. Furthermore, the best-fit packing method does not prevent tokens from different documents from appearing in training sequences, whereas our method does. The presence of padding tokens in best-fit packed sequences also means that some context is wasted during each optimization step.

## E  Training stability with VSL and curriculum

[31] presents the *stability-efficiency dilemma*: efficient LLM pretraining with massive data parallelism results in a large batch size and requires a high learning rate. However, such a setup can result in training instability, leading to poor generalization. They observe a correlation between training instability and long sequences, especially at the early stages of training, suggesting that training on long sequences when the model is not well-trained can be a main source of training instability.

Here, we show that dataset decomposition alleviates this problem when used with a curriculum: starting training by sampling more from short sequence buckets. We empirically demonstrate this by training an OpenLM-1B model from scratch with a high learning rate ($= 10^{-2}$) and no gradient clipping, once with baseline-8k and once with DD using the "Grow-P100" curriculum. Training loss is shown in Fig. 8, demonstrating the stability of training with DD in comparison to the baseline. This suggests that our proposed method can also be beneficial for large-scale pretraining with large batches and high learning rates in terms of efficiency.

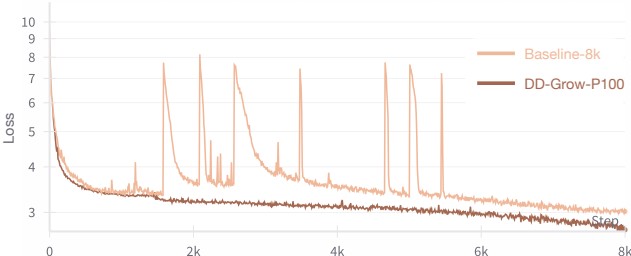

Figure 8: We compare the training loss when training with Baseline-8k versus DD with the "Grow-P100" curriculum. Both models are trained with identical hyperparameters, a high learning rate ($= 10^{-2}$), and no gradient clipping. It is evident that DD results in greater stability.

## F  Average sequence length vs average context length

We compute the mean of length (Fig. 3a) and context (Fig. 3c) distributions as follows. Assume a list of sequences with lengths $l_1, l_2, \ldots, l_N$, which are, for example, the chunk lengths in the concat-and-chunk approach or the sequence lengths in different buckets of the dataset decomposition approach. We define the average sequence length as follows:

$$\text{Average sequence length} = \frac{1}{N} \sum_i^N l_i \tag{1}$$

In auto-regressive training on a sequence with length $l$, we apply $l$ losses for next-token prediction on each token in parallel. Hence, for a sequence with length $l$, we see contexts with lengths equal to

$0, 1, 2, \ldots, l-1$. We define the average context length, which is different from the average sequence length, as follows:

$$\text{Average context length} = \left( \sum_{i=1}^{N} \sum_{j=0}^{l_i-1} j \right) \Big/ \left( \sum_{i=1}^{N} l_i \right) = \left( \sum_{i=1}^{N} l_i(l_i-1) \right) \Big/ \left( 2 \sum_{i=1}^{N} l_i \right). \quad (2)$$

In Fig. 3a, Fig. 3c, and Table 1, we report the average sequence length and average context length for original documents, concat-and-chunk, and dataset decomposition with different mixtures.

