# OpenReview forum: "Dataset Decomposition: Faster LLM Training with Variable Sequence Length Curriculum"
_NeurIPS.cc/2024/Conference — NeurIPS 2024 poster_

### Official Review · Reviewer_uip1 · 2024-07-06

**Soundness:** 3
**Presentation:** 3
**Contribution:** 3
**Rating:** 5
**Confidence:** 4

**Summary:**

This paper proposes to carefully segment the training data so that different documents won't be mixed together. They also propose a Grow-P2 curriculum that increases training efficiency and stability.

**Strengths:**

The proposed Grow-P2 curriculum is useful to practitioners if they want to pretrain large language models.

**Weaknesses:**

This paper presents only empirical results.

**Questions:**

1. The authors only explore the training curriculum regarding lengths. There are other dimensions when it comes to curriculum design. For example, should we train the model on simpler topics first?

**Limitations:**

None.

---

> ### Author Rebuttal · Authors · 2024-08-06
>
> We would like to thank reviewer uip1 for their feedback. We respond to the reviewer’s concerns and questions below and kindly request that you let us know if further clarification is needed.
>
> ---
> > This paper presents only empirical results.
>
> We appreciate that our work is considered an empirical contribution, yet valuable to the community. We respectfully do not consider this a weakness.
>
> ---
> > other dimensions for curriculum
>
> As suggested by the reviewer, different notions of difficulty can be explored when designing a curriculum. In this work, our focus is on sequence length-based curriculum. We show that such a curriculum results in both training efficiency (through variable sequence length training) and performance gain.

---

> > ### Comment · Reviewer_uip1 · 2024-08-10
> > **I have read the rebuttal.**
> >
> > Thank you for providing additional details. I'll keep my positive score at 5.

---

> > > ### Author Response · Authors · 2024-08-13
> > > **Appreciate your response**
> > >
> > > We would like to thank Reviewer *uip1* again for their positive feedback.

---

### Official Review · Reviewer_H2jF · 2024-07-12

**Soundness:** 3
**Presentation:** 2
**Contribution:** 3
**Rating:** 6
**Confidence:** 3

**Summary:**

The paper explores dataset decomposition for LLM pre-training.

The method decomposes documents into subsequences and organizes them into buckets. Sequences of similar lengths are grouped in the same bucket, and different buckets have different lengths. This amounts to more efficient training. The paper investigates various mixtures of lengths separately for their impact on performance.

The paper also explores a length-based cyclic curriculum learning - treating smaller length buckets as "easy" examples and larger ones as "hard" examples. Curriculum learning can improve the results a bit.

**Strengths:**

* The high-level ideas are reasonably motivated.
* Reasonably extensive experimental analyses of the methods are provided along with baseline dataset structuring methods for pre-training.
* Results are generally promising. Generally, it results in better task performance while taking less training time.

**Weaknesses:**

On the one hand, the proposals can be seen as important in exploring some unorthodox training structures for the specific context of LLMs and informing future pre-training. However, on the other hand, the paper seems to be mainly an exploration of hyperparameter tuning. The main proposed techniques seem like extensions of existing strategies (bucketing and curriculum learning) for LLMs. Bucketing is already understood to make things efficient, and curriculum learning has some positive results in NLP in general in earlier papers. Cyclic curriculum learning was also used in earlier works [1].

[1] CYCLICAL CURRICULUM LEARNING - Kesgin et al. ArXiv 2022

Also, if I understand correctly, the training speed gain may be less significant with flash-attention-based LLMs and alternative models like (Mamba, Linear Transformers).

**Questions:**

Question:

1. Can cross-document attention be restricted with an attention mask? Can that be explored?
2. Wouldn't a cyclic curriculum still be a problem if the learning rate starts to degrade before the first curriculum cycle? If I understand correctly, you have to run the cycles quicker than the learning rate decay? More discussion on this can be helpful.
3. Can you elaborate more on the pacing strategy (see more on my confusion below in the suggestions)?


Suggestions:

* I did not quite understand the exact details of the pacing mechanism for the cyclic curriculum. It would be helpful to provide a pseudo-code or explain it through mathematical formalisms. The table shows a static distribution of sampling odds, but if the model is shifting from easy to hard examples, then the sampling odds should be changing, no? It's not clear to me how exactly that is being done.

**Limitations:**

Yes.

---

> ### Author Rebuttal · Authors · 2024-08-06
>
> We would like to thank the reviewer H2jF for their time and feedback. We are glad that the reviewer finds our work to be reasonably motivated, and the results to be extensive and promising. We respond to the reviewer’s concerns and questions below, address all of them in the revised paper, and kindly request that you let us know if further clarification is needed.
>
> ---
> > The paper seems to be mainly an exploration of hyperparameter tuning
>
> We respectfully disagree with this comment. We do not tune any hyperparameters in this work. The dataset (RefinedWeb), models, and training hyperparameters are all based on publicly available OpenLM repo, and are used for both the baselines and our method without any tuning.
>
> ---
> > The main proposed techniques seem like extensions of existing strategies
>
> While some components introduced in this work (multi-stage training based on sequence length, length-based bucketization, and cyclic schedules) have been used by previous works in different contexts, this is **the first work to show the efficacy of length-based curriculum for autoregressive LLM pre-training**, both in terms of training efficiency (faster training) and training performance (final model accuracy, especially for long-context metrics). We would like to emphasize that given the significant cost associated with LLM pretraining, the savings provided by the proposed method are very substantial (up to 45% faster training for models considered in the paper).
>
> In addition, while our bucketing and curriculum (e.g., [1]) may look similar to prior works for different domains and setups, they are novel and have major differences as explained below:
>
> * **Bucketing**: We introduce binary decomposition (Section 2.1), a novel method to preserve document length, form buckets with fixed sequence lengths without using pad tokens, and avoid forming multi-document sequences (thus achieving no cross-document attention without attention masking). This is different from existing bucketing methods in different domains, where cross-document attention may still occur (albeit with reduced chances) without padding.
> * **Sequence length-based curriculum**: Our analysis shows the importance of mixing sequences with different lengths during training and curriculum (Table 2), beyond the simple multi-stage training considered by previous works.
>
> Please also see [our response to reviewer bcif on the novelty](https://openreview.net/forum?id=r8M9SfYMDi&noteId=TtzwitydGc).
>
> [1] CYCLICAL CURRICULUM LEARNING - Kesgin et al. ArXiv 2022
>
> ---
> > The training speed gain may be less significant with flash-attention-based LLMs and alternative models
>
> All results presented in this paper are with FlashAttentionV2, and we achieve up to 45% faster training and more accurate models (see Fig 1b) compared to the baseline, which also uses FlashAttentionV2 for sequence length up to 8192. Training speed gains will be even more when training on longer sequences.
>
> Please note that the proposed method is designed to speed up the training of transformers and further increase their performance without any architectural changes. Alternative architectures, such as state-space based models (e.g., Mamba) or approximations to attention (e.g., linear attention), do not suffer from quadratic attention costs but come with performance limitations. We would like to emphasize that transformers with multi-head attention are still the predominant architecture in most large-scale language models in the community.
>
> ---
> > Can cross-document attention be restricted with an attention mask? Can that be explored?
>
> Yes, we have already included results with attention masking to avoid cross-document attention. Baseline-8k-DM and Pack-8k+DM in Table 5 refer to models trained with Document-Masking (DM). Applying document masking mainly improves the regular evaluation, making it closer to our results. However, document masking does not provide the computational benefits (training speed) of our proposed method. Please **see Tables 1 and 2 in the rebuttal PDF**.
>
> ---
> > Cyclic curriculum vs learning rate schedule
>
> In all our experiments, we use a learning rate schedule with a short initial warmup followed by a one-cycle cosine learning rate decay, as in the OpenLM repository. For the length curriculum, we analyze both one-cycle and multi-cycle curricula. In all cases, the warmup period for the learning schedule is shorter than a single length curriculum. We did not observe any stability or convergence problems in any of the setups. In fact, in Appendix E, we show that our proposed curriculum significantly improves training stability compared to the baseline. To further clarify our cyclic length curricula with respect to the learning rate schedule, please **see Figure 2 in the rebuttal PDF** for a visualization overlapping both schedules (to be also included in the revised paper).
>
> ---
> > Details of the pacing mechanism
>
> We apologize for the confusion. We provide further clarification on our mixture and curriculum implementation below with a pseudo-code, as suggested (to be also included in the revised paper). We will also **release the full code** (a small patch on top of the OpenLM repo) which should further clarify the details.
>
> ```
> # {D_i}: list of buckets such that D_i includes sequences with length 2^i
> # {n_i}: total number of tokens to be picked from each bucket (see Table 1 of paper)
> # {o_i}: sampling odd for each bucket (see Table 2 of paper)
> # c: number of cycles
> # b: number of tokens per optimization step
>
> # Form c non-overlapping random subsets from each bucket D_i:
> s_{i,j} = random subset of D_i with n_i/c tokens # for j = 1, 2, ..., c
>
> for j in [1, 2, ..., c]: # loop over cycles
>     while at least one s_{i,j} is not empty:
>         odds = [o_i if s_{i,j} not empty else 0 for i=1,2,3,...]
>         probs = odds / odds.sum()
>         randomly sample index i with probability probs[i]
>         sample b/2^i sequences from s_{i,j} without replacement and use for training
> ```

---

> ### Comment · Reviewer_H2jF · 2024-08-08
> **Response**
>
> Thank you for the rebuttal. Overall, I increased the score to 6.
>
> > We respectfully disagree with this comment. We do not tune any hyperparameters in this work. The dataset (RefinedWeb), models, and training hyperparameters are all based on publicly available OpenLM repo, and are used for both the baselines and our method without any tuning.
>
> I meant in the sense that training scheduling strategies and data sampling can be seen as a form of hyperparameter, and exploring small changes in the details of established strategies (bucketing, curriculum learning) can be in itself seen as hyperparameter tuning. I am not saying that there is any technical issue here, just that technical novelty may appear limited as a result.
>
> > All results presented in this paper are with FlashAttentionV2, and we achieve up to 45% faster training and more accurate models (see Fig 1b) compared to the baseline, which also uses FlashAttentionV2 for sequence length up to 8192. Training speed gains will be even more when training on longer sequences.
>
> Thank you for the clarification.
>
> > In all our experiments, we use a learning rate schedule with a short initial warmup followed by a one-cycle cosine learning rate decay, as in the OpenLM repository. For the length curriculum, we analyze both one-cycle and multi-cycle curricula. In all cases, the warmup period for the learning schedule is shorter than a single length curriculum. We did not observe any stability or convergence problems in any of the setups. In fact, in Appendix E, we show that our proposed curriculum significantly improves training stability compared to the baseline. To further clarify our cyclic length curricula with respect to the learning rate schedule, please see Figure 2 in the rebuttal PDF for a visualization overlapping both schedules (to be also included in the revised paper).
>
> My point is more related to the theoretical motivation you supplied for cyclic curriculum - that due to learning rate decay, by the time the model starts encountering hard examples.
>
> "Due to the presence of other hyperparameter schedules during the course of training (e.g., learning
> 259 rate and weight decay), a curriculum on length may result in a potential implicit bias. For example, if
> 260 we only see long sequences toward the end of training, long sequence learning occurs only when the
> 261 learning rate is too small. To address this potential issue, we also explore cyclic curricula, where a
> 262 curriculum is applied in cycles (similar to cyclic learning rate schedules [52])."
>
> My point is that the cyclic curriculum itself doesn't seem to be a complete solution, but it has to be balanced properly with learning rate scheduling if this motivation applies at all. Following this motivation, if you have a monotonic decay of learning rates , and it's faster than the first cycle - the same problem arises. Also, on the other hand, if you have cyclic learning rates, the theoretical motivation for a cyclic curriculum seems to diminish because it can allow harder examples to experience higher learning rates near the end without cycles.
>
> More can be discussed here in these regards.
>
> > We apologize for the confusion. We provide further clarification on our mixture and curriculum implementation below with a pseudo-code, as suggested (to be also included in the revised paper). We will also release the full code (a small patch on top of the OpenLM repo), which should clarify the details further.
>
> Thank you. That brought some much-needed clarity on the method.
>
> It seems you don't strictly have any explicit pacing function to directly control the odds of sampling easier samples (starting from high odds) as time goes on. Instead, you do sampling without replacement. So once the easy samples are made early on, near the end, mostly harder samples will remain to sample from. I think this should be discussed more in the text.

---

> > ### Author Response · Authors · 2024-08-09
> > **Appreciate your response**
> >
> > We would like to thank Reviewer H2jF again for their time and positive feedback.
> >
> > As suggested by the reviewer, we will include more discussion in the revision on our investigation of length curriculum in relation to other schedules, such as learning rates.
> >
> > Regarding the method details: The reviewer’s understanding is correct. We do not explicitly control the pace; instead, we sample without replacement using fixed probabilities (that determine the curriculum). When a bucket is empty for the current cycle, it is excluded from sampling. Consequently, as the reviewer mentioned, mostly long sequences remain toward the end. Note that even when we train on a sequence of length $n$, we have $n$ next-token prediction losses (applied in parallel) with context lengths $0, 1, 2, …, n-1$. This implies some mixing: when training on a hard example (i.e., a long sequence), we also have easy examples (its shorter sub-sequences). Therefore, even toward the end of each cycle, we still have some losses with short contexts. As suggested, we will augment the text with additional discussion in the revision.

---

### Official Review · Reviewer_z2w6 · 2024-07-12

**Soundness:** 2
**Presentation:** 3
**Contribution:** 3
**Rating:** 5
**Confidence:** 3

**Summary:**

The paper introduces 'Dataset Decomposition' (DD), a method to enhance the pre-training of Large Language Models (LLMs). Contrary to the traditional 'concat-and-chunk' approach, which can lead to unwanted cross-document attention and computational inefficiency, DD organizes datasets into buckets with sequences of uniform length from individual documents, enabling variable sequence length training. This method is demonstrated to reduce training time, improve model performance, and scale effectively with the size of the dataset.

**Strengths:**

1. Comprehensive Experiments: The authors conduct extensive experiments across various datasets and LLMs, with observations and analyses that are noteworthy.

2. Simplicity and Effectiveness: The proposed method is straightforward and efficacious.

**Weaknesses:**

The paper's writing requires improvement, as exemplified by the following:

Line 14: "Our proposed method incurs a penalty proportional to the actual document lengths at each step, resulting in significant savings in training time." Even after careful reading, it remains unclear what is meant by "penalty" here.

Line 158 (and many other instances): "We follow the exact setup as in [32]." When using citations within sentences, it is customary to use the \citet command, resulting in "We follow the exact setup as in Liu et al. (2024)," rather than using \cite or \citep.
Table 1: It would be beneficial to clearly define the meanings of the elements in the first column within the main body of the paper to prevent confusion.

Additionally, while it adds value to the paper to provide extensive discussions of various aspects through experiments, the overall organization of the paper should be enhanced to make the main focus of the paper clearer.

In Section 3.6, DD shows a significant advantage over other methods in long-context scenarios. However, the authors do not offer a detailed analysis of why this is the case. In my understanding, with a fixed number of training tokens per gradient update, DD increases efficiency by enlarging the batch size and reducing sequence length (<8k). In contrast, the baseline and ICLM maintain an 8k sequence length. It is unclear why DD performs better, as most of its training samples are actually shorter in length.

**Questions:**

The same as the weaknesses

**Limitations:**

No potential negative societal impact.

---

> ### Author Rebuttal · Authors · 2024-08-06
>
> We would like to thank reviewer z2w6 for their time and feedback. We are happy that the reviewer finds our experiments comprehensive, analyses noteworthy, and the method simple and effective. We address the concerns and questions raised by the reviewer below and kindly request to be informed if further clarification is needed.
>
> ---
> > Paper's writing requires improvement
>
> We thank the reviewer for the editorial points. We have applied all of them in the revised manuscript including additional discussion and apologize for any potential inconvenience during the review.
>
> ---
> > It remains unclear what is meant by penalty
>
> We apologize for the confusion. By "penalty," we meant "computational cost." We fix this in the revised manuscript.
>
> ---
> > Significant advantage over other methods in long-context scenarios
>
> We provide a more detailed discussion here, and revise the paper accordingly.
>
> As pointed out by the reviewer, pre-training context length is an important factor in determining a model’s long-context performance. We empirically validate this in the results shown in Fig. 5a of the paper, where models trained on longer sequences perform better on multi-document QA. For the context length from the same document (i.e., the number of tokens from the same document a token can attend to), our proposed method has an average context length of 1,344 for the RefinedWeb dataset (as defined in equation 2 in Appendix F), compared to 930 for the baseline (see Figure 3c of the paper) and 1,064 when bin-packing [1] is applied. This explains why the dataset decomposition mixture, even without any length-based curriculum (the first row in Table 2 of the paper), outperforms Baseline-8k-DM and Pack-8k+DM (second and third rows in Table 5 of the paper). Here, DM refers to applying document masking to avoid cross-document attention.
>
> One can increase context length by concatenating different documents and putting them in the context, as in the Baseline-8k result. However, simply increasing the context length by filling it with multiple documents does not necessarily lead to long-range attention during training (and hence improved long-context capability of the model). Nevertheless, as discussed in the response to reviewer bcif, a multi-document context encourages the model to learn to discern and disregard irrelevant information. Comparing Baseline-8k and Baseline-8k-DM multi-document QA results in Table 5 of the paper shows such benefit. Baseline-8k multi-document QA performance is even slightly better than our proposed dataset decomposition mixture when used without length-based curriculum (first row of Table 2 of the paper).
>
> In-context pre-training LMs (ICLM [2]) proposes to put semantically relevant documents into the context. We observe that ICLM results in slightly better multi-document QA performance when 30 documents are in the context compared with Baseline-8k (22.0% vs. 20.5%). However, we do not observe such gains in shorter multi-document QAs (i.e., with fewer distractor documents in the context) and regular evaluations.
>
> Finally, in Table 2 of the paper, we show the importance of length-based curriculum. Note that the data mixture (and hence the average context length) is the same for all rows in Table 2 of the paper, differing only by the curriculum (i.e., the order in which different examples are seen during training). We show that using our proposed cyclic length-based curriculum, for example, Grow-P2 with 8 cycles, results in a significant improvement in the model’s long-context capability. For instance, multi-document QA with 30 documents improves from 19.6% with no curriculum (first row in Table 2 of the paper) to 24.6%. It is worth noting that the effect of length-based curriculum on regular metrics is less significant, with the average metric improving from 53.8% with no curriculum to 54.4% for Grow-P2 with 8 cycles curriculum.
>
> Please **see Table 1 of the rebuttal PDF** for a  summary of all of the above contributing factors.
>
> [1] Fewer truncations improve language modeling, ICML 2024
>
> [2] In-context pretraining: Language modeling beyond document boundaries, ICLR 2024

---

> > ### Comment · Reviewer_z2w6 · 2024-08-14
> > **Appreciate your response**
> >
> > Thank you for your detailed response. I'm glad to see that the writing has been improved. I have changed the score accordingly.

---

> > > ### Author Response · Authors · 2024-08-14
> > > **Appreciate your feedback**
> > >
> > > We would like to thank the Reviewer *z2w6* for their positive response.

---

### Official Review · Reviewer_bcif · 2024-07-15

**Soundness:** 2
**Presentation:** 3
**Contribution:** 3
**Rating:** 5
**Confidence:** 4

**Summary:**

The paper introduces a method called dataset decomposition for training large language models (LLMs) more efficiently. Traditional LLM training processes use fixed-length token sequences, leading to inefficiencies such as unnecessary computational costs from cross-document attention. The proposed method tackles this by organizing the training dataset into various "buckets," each containing sequences of a fixed length from a unique document. This allows for variable sequence length training, where different buckets can be sampled during training based on a curriculum that adjusts for sequence length. The approach significantly reduces the attention computation overhead, leading to faster training times and improved model performance across various language understanding benchmarks. This method enables efficient and scalable LLM pretraining on large datasets, with experimental results showing up to three times faster attainment of target accuracies compared to traditional methods.

**Strengths:**

1: The paper is well organized and easy to follow.

2: The motivation is clear and the proposed method looks simple yet effective.

3: This paper conducted massive experiments and provided many valuable empirical results, which can support both the claim of this paper as well as many long existing guesses in the community.

**Weaknesses:**

1. It might be more accurate to consider that "the cross-document attention allocates significant computational resources to attending to unrelated tokens that may not directly contribute to learning." However, it's also valuable for models to develop the ability to discern and disregard irrelevant information.

2. The concept of a length-based curriculum, while insightful, isn't entirely novel. It has been explored in previous studies, such as those detailed in references [1] and [2]. Furthermore, numerous models, including BERT, employ similar curriculum learning strategies, albeit without specific emphasis on this aspect.

[1] World Model on Million-Length Video and Language with RingAttention
[2] GrowLength: Accelerating LLMs Pretraining by Progressively Growing Training Length

**Questions:**

1. How can we ascertain the outcomes when models are trained with sufficient data? Will there remain a gap between the proposed method and the baseline?

In a short summary, my major concern is the originality of the proposed strategy. Nonetheless, the empirical findings presented in this paper certainly add valuable contributions to the community.

**Limitations:**

Yes

---

> ### Author Rebuttal · Authors · 2024-08-06
>
> We would like to thank the reviewer, bcif, for their time and feedback. We are glad the reviewer finds the paper well-organized and motivated, the method effective, and the experiments valuable to the community. In the following, we address the concerns and questions raised by the reviewer and kindly request to be informed if further clarification is needed.
>
> ---
> > It's also valuable for models to develop the ability to discern and disregard irrelevant information
>
> We agree with the reviewer that, in principle, the model may benefit from attending to irrelevant documents during pre-training to develop a discerning capability. We mentioned this in the paper (line 291). Our results also confirm this. In Table 5, when comparing baseline-8k to baseline-8k with document masking (which stops cross-document attention), adding document masking improves the regular metric (51.5% → 52.4%). However, it results in weaker discerning capability, as seen in multi-document QA results with 30 documents in the context (performance drops from 20.5% → 16%). We will clarify this point further in the revision.
>
> We would also like to point out that our proposed method surpasses all baselines in discerning capability, as seen from its superior performance on multi-document QA. This capability emerges when a length-based curriculum is deployed (compare the multi-document QA performance with a uniform curriculum and, for example, the Grow-P2 curriculum in Table 2). Please **see the summary in Table 1 of the rebuttal PDF**.
>
> ---
> > Length-based curriculum, while insightful, isn't entirely novel
>
> We thank the reviewer for pointing out two relevant works. Both are **concurrent** with ours **and not published** in a peer-reviewed venue at the time of this rebuttal. We also appreciate the reference to the length-based curriculum in BERT [3]. We will include them in the revised paper. These works highlight the computational benefits of length-based batching; however, they differ significantly from our work, as explained below:
>
> * We are the first to show length-based curriculum for autoregressive LLM pre-training. Given the huge cost of LLM pre-training, the savings from our proposed method are significant (more than 6x speed up to reach the best accuracy of baseline as shown in Figure 1 of the rebuttal PDF). The related work [1] is a continual learning setup (starting from pre-trained Llama2) only for context-length extension using book data. GrowLength [2] does not show any results on large language models. BERT [3] is only for masked-language modeling.
> * Unlike [1-3], we introduce binary decomposition, a novel method to preserve document length, form buckets with fixed sequence lengths without using pad tokens, and avoid forming multi-document sequences (thus achieving no cross-document attention without attention masking).
> * Unlike [1-3], we analyze different forms of curriculum (not just a simple multi-stage training from short to long sequences) in Table 2, and show that mixing (i.e., a mixture of long and short sequences with a changing mixture) is important for the best results. Our analysis further shows that our introduced cyclic schedule is important for best results.
> * Unlike [1-3], we systematically show the effect of pre-training sequence length on model performance for different tasks, including those requiring long context (Sections 3.2 and 3.3).
>
>
> [1] World Model on Million-Length Video and Language with RingAttention, 2024
>
> [2] GrowLength: Accelerating LLMs Pretraining by Progressively Growing Training Length, 2024
>
> [3] Pre-training a BERT with curriculum learning by increasing block-size of input text, RANLP 2021
>
> ---
> > The outcomes when models are trained with sufficient data
>
> Note that the benefits of using the proposed method are twofold:
>
> * **Computational benefit**: Using variable sequence length training (Section 2.2), the training cost per seen token is reduced, i.e., we reach a certain performance faster compared to the baseline.
> * **Performance benefit**: Using DD with a curriculum for the same total number of seen tokens, compared to the baseline, DD results in a more accurate model (for both regular and long context evaluations).
>
> To further demonstrate the above facts and the scalability of our results, we trained a 410M model with our proposed method and the baseline up to 1.1 trillion tokens. This is 128 times more tokens than the “optimal” number recommended based on the number of parameters of the model in Chinchilla paper [4]. Furthermore, 1.1 trillion tokens exceed the number used in recent state-of-the-art open LLMs (e.g., smolLM [5] uses 600 billion tokens to train their 360M model). We show **+2.4 accuracy improvement compared to baseline even at 1.1 trillion tokens**, where the baseline show a plateau in accuracy (indicative of sufficient data). Further we show **more than 4x data efficiency** and **more than 6x speed-up** to reach to baseline's best accuracy. Please **see Figure 1 of the rebuttal PDF**.
>
> [4] Training compute-optimal large language models, NeurIPS 2022
>
> [5] SmolLM - blazingly fast and remarkably powerful, 2024
>
> ---
> > The originality of the proposed strategy
>
> We would like to emphasize again that this is the first work demonstrating the computational and performance benefits of length-based curricula (and beyond simple multi-stage training) for LLM pre-training. Furthermore, the proposed binary dataset decomposition method is entirely new. As mentioned by the reviewer, we provide extensive experiments and insights on the importance of length in the mixture, training stability, and generalization to different model sizes and datasets. We believe the combination of our proposed binary decomposition method, extensive empirical results, and open-source code and models will be a useful contribution to the community.

---

> > ### Comment · Reviewer_bcif · 2024-08-13
> >
> > Thanks for the response. It solved my concerns. Although, the novelty is still questionable. I believe some empirical results from this paper is valuable to the community. Hope more discussion about those related works can be added to the revised version.
> >
> > I raised my score to 5.

---

> > > ### Author Response · Authors · 2024-08-13
> > > **Appreciate your response**
> > >
> > > We are glad that Reviewer *bcif*’s concerns have been addressed and would like to thank them again for their positive feedback.

---

### Author Rebuttal · Authors · 2024-08-06

We would like to thank all the reviewers for their time and feedback. We are pleased with the positive comments from the reviewers: Reviewer *bcif* finds the paper **well-organized and motivated**, the **method effective**, and the **experiments valuable to the community**; Reviewer *z2w6* finds our **experiments comprehensive**, our **analyses noteworthy**, and the **method simple and effective**; Reviewer *H2jF* finds our work **reasonably motivated**, and the **results extensive and promising**, and Reviewer *uip1* find our work **useful to practitioners**.

We provide multiple tables and figures in the **one-page PDF rebuttal** to further support the contributions of the paper and clarify questions and concerns raised by the reviewers. We show **more than 4x data efficiency** and up to **more than 6x training speed up** compared to the baseline for **large-scale trainings up to 1.1 trillion tokens**. We would like to emphasize that all contributions of this paper, including code and all model checkpoints, will be **open-sourced** to facilitate follow up works. We address individual reviews below and kindly request that you let us know if any further questions or concerns remain unaddressed.

---

### Decision · Program_Chairs · 2024-09-25

**Decision:**

Accept (poster)

**Comment:**

This paper proposes a data strategy for LLM pre-training, specifically, it buckets training sequences into variable lengths and then sample a batch from one bucket based on some pre-defined curriculum learning during training. This training data strategy can reduce training overhead as shown in the paper and speed up training of LLMs effectively.

Although some reviewers question about the novelty and originality of the idea as there are already public research showing the effectiveness of multi-stage pre-training on documents of increasing sequence lengths, I found the experiments and analysis conducted in this paper quite insightful for LLM training community.

One issue authors should pay attention to is one of their motivations in the paper "this method of concatenation can lead to cross-document attention within a sequence, which is neither a desirable learning signal nor computationally efficient." Actually, modern data concatenation approach already eliminates cross-document attention by specifying document boundaries in efficient attention implementations like FlashAttention2. Thus, I think this is not a valid point in claiming the advantages of the proposed method.

Overall, I recommend acceptance of this paper as it presents a clear, simple and effective approach with solid empirical results.